# The Association between Body Mass Index and Muscular Fitness in Chinese College Freshmen

**DOI:** 10.3390/ijerph192114060

**Published:** 2022-10-28

**Authors:** Feng Sun, Qiang He, Xiaohan Sun, Jianxin Wang

**Affiliations:** 1Institute of Sports Science College, Nantong University, Nantong 226019, China; 2College of Physical Education, Shandong University, Jinan 250061, China

**Keywords:** body mass index, muscular fitness, college freshmen, physical fitness

## Abstract

(1) Background: The present study aimed to investigate the association between body mass index (BMI) and muscular fitness in Chinese college freshmen. (2) Methods: A total of 6425 college freshmen in mainland China were recruited. BMI was classified as underweight (<18.5 kg/m^2^), normal weight (18.5~23.9 kg/m^2^), overweight (24~27.9 kg/m^2^), and obese (≥28 kg/m^2^), according to the Working Group on Obesity in China. Health-related physical fitness components including cardiorespiratory fitness, lower body explosive power, upper body muscular endurance, abdominal muscular endurance, flexibility, and vital capacity were assessed. Physical fitness index and muscular fitness index were calculated, respectively, as the sum score of the standardized values (z-score) of the corresponding components. Three regression models were used to evaluate the potential associations: a linear regression model, a polynomial regression model, and a restricted cubic spline regression model. Adjust R square was used to compare among models. (3) Results: Significant differences were observed among different BMI categories in nearly all physical fitness components as well as physical fitness z-score and muscular fitness z-score (*p* < 0.001), regardless of gender. Significant linear associations were found between BMI and physical fitness z-score as well as muscular fitness z-score among total, male, and female groups (*p* < 0.05). However, the restricted cubic spline regression model showed a better fitting effect (adjust R^2^ was 7.9%, 11.2%, and 4.8% in total, male, and female for physical fitness and 7.7%, 15.7%, and 4.0%, for muscular fitness, respectively), compared with the linear and polynomial regression models, presented by a higher adjusted R^2^. Restricted cubic splines analysis showed that BMI value and physical fitness z-score showed a non-linear relationship with an approximate inverted U curve in all groups, while an approximate reversed J-shaped association was observed between BMI and muscular fitness z-score in all groups. (4) Conclusions: The present study showed a nonlinear negative relationship between BMI and physical fitness with underweight and overweight/obese college freshmen having poorer physical fitness and muscular fitness than their normal BMI peers, which may provide useful evidence to the development of public health recommendations and encourage the health management of young adults. Future studies should further explore the relationship between BMI and muscular fitness with multi-centered large sample size studies.

## 1. Introduction

Physical fitness is a crucial marker of health. It can be defined as an integrated measure of various body functions which correspond to several performances of daily physical activity and/or exercise [1]. Physical fitness consists of various components, including cardiorespiratory fitness, muscular strength, flexibility, etc. The role of cardiorespiratory fitness and health outcomes have been extensively studied in the past decades. Evidence has strongly supported that cardiorespiratory fitness is an independent predictor of all-cause and cardiovascular disease (CVD) mortality [2,3]. People with lower levels of cardiorespiratory fitness are associated with a higher relative risk of all-cause mortality (RR = 1.7) and CVD events (RR = 1.56) [3]. Recently, the importance of muscular fitness to health has become widely recognized. Muscular fitness is defined as an incorporation of manifestations of muscle strength, including maximal strength (isometric and dynamic), explosive strength, endurance strength, and isokinetic strength [4]. Higher level of muscle strength was associated with a significant reduced risk of all-cause mortality (HR = 0.69) [4] and lower metabolic risk [5]. In addition, there have also been increasing investigations into physical fitness and obesity. For instance, it has been found that children and adolescents with high cardiorespiratory fitness have significantly lower total adiposity regardless of fatness measured by skinfold thicknesses [6], dual energy X-ray absorptiometry, or magnetic resonance imaging [7]. Similar results were observed in adults [8]. In addition, it seems that people with higher muscular strength have lower risk of overweight or obesity [9].

Although partly genetically determined, physical fitness can be influenced by a series of factors, particularly for physical activity, sedentary behavior, and obesity. Reduced physical activity and increased sedentary time pose a risk of lower level of physical fitness as well as increased risk of chronic conditions [10]. College students constitute large proportion of young adults, over 35% in most developed countries. Research has indicated that a large proportion of college students fail to meet the guidelines for physical activity published by authoritative organizations [11]. A national Chinese survey showed a high prevalence of physical activity times of less than 1 h per day in students aged 9–22 years, and the highest prevalence was in 18-year-old male students (82.5%) and 21-year-old females (89.8%) [12]. In addition, college students spend 7.29 h per day (self-reported) and 9.82 h per day (measured by accelerometers) in sedentary behavior, which is associated with activities requiring long periods of sitting like lectures, studying in the library, screen time, etc. [13]. Accumulations of sedentary behavior are associated with higher body mass index (BMI) and waist circumference (WC) [14]. Physical fitness is also affected by a variety of other factors; among these, obesity plays a vital role [15,16]. People with higher body fat usually presented lower cardiorespiratory and muscular endurance fitness [17,18]. The prevalence of obesity has strikingly increased over the past few decades and become a public health threat all over the world [19]. The rapid development of China witnessed a rising prevalence of obesity and recent data demonstrated that the prevalence of overweight and obesity in adults aged over 18 reached 34.3% and 16.4%, respectively [20]. It is estimated that the prevalence of overweight and obesity might reach 65.3% in adults by 2030 [21]. In recent years, the prevalence of overweight and obesity in young adults, especially college students, has increased rapidly in many countries [22]. However, relatively less studies explored the association between obesity and physical fitness in college student populations, particularly for muscular fitness. In addition, most current studies assume a linear relationship between BMI and physical fitness, and thus analyzed these by linear regression model or simple person correlation analysis [23,24]. However, these studies did not fully consider BMI variability, which was usually categorized into normal weight, overweight, and obesity groups. This might be associated with more overweight and obesity samples being included in these studies. Consequently, most existing studies reported a reverse J-shaped curve in terms of the relationship between BMI values and physical fitness. However, underweight represented another extremity of weight status which should be also considered.

Therefore, although numerous studies have investigated the relationship between weight status and physical fitness, few studies examined it in college freshmen population. This study aims to evaluate the association between weight status and physical fitness, especially for muscular fitness in Chinese college freshmen by different analysis models.

## 2. Materials and Methods

### 2.1. Participants

The cross-sectional data were derived from the annual physical fitness test of college freshmen 2017–2020 in Nantong University, Jiangsu Province, China. All freshmen were enrolled in the test in their first year of university (2017, 2018, 2019, and 2020). Disabled students or those who were sick during the test were allowed to skip the test if they provided valid supporting documents. All tests were conducted from October to November in 2017, 2018, 2019, and 2020. A total of 7980 students took the test, while the data of 1555 students were incomplete. Finally, a total of 6425 college students (15 to 23 years old in their first college year) were finally included for analysis. Written informed consent was obtained from students or their parents/guardians (≤16 years). This cross-sectional study was approved by the Ethics Committee of Nantong University (Ref. No: 2018–K037).,

### 2.2. Anthropometric Measurements

Anthropometric measurements were conducted by well-trained staff. Height and weight were recorded with BMI calculated as weight (kg)/height (m^2^). BMI category were generally classified into underweight (<18.5 kg/m^2^), normal weight (18.5~23.9 kg/m^2^), overweight (24~27.9 kg/m^2^), and obese (≥28 kg/m^2^) in accordance with the criteria of the Working Group on Obesity in China.

### 2.3. Physical Fitness Assessment

This study used six fitness items to assess the physical fitness, including vital capacity weight index (vital capacity adjusted by weight), speed (50 m sprint), low body explosive power (standing long jump), flexibility (sit and reach), cardiorespiratory endurance (800 m/1000 m run for girls and boys, respectively), abdominal muscular endurance (sit-ups), and upper body muscular endurance (pull-ups). Vital capacity was measured by a calibrated electronic spirometer (HK6800-FH, Hongkangjiaye, China) and scored as the maximal volume of air (mL) that each participant could expel from their lungs after a maximal inhalation (measured twice). Vital capacity weight index (mL/kg) was then calculated as vital capacity (mL) divided by weight (kg). Speed was determined by the time taken to finish a 50-m sprint test (test only once). Participants were instructed to run as fast as they could for 50 m along a straight line on the athletic track. The sit-and-reach test required each participant to gradually reach forward with the fingers as far as possible and was scored as the most distant point on the ruler (the better record of two trials). The standing long jump was measured by the distance from the starting line to the heel of the closet foot when participants were asked to jump forward as far as possible (measured twice). Male and female participants were instructed to complete the pull-ups/sit-ups test as many times as possible in 60 s, respectively. Male students were instructed to grasp an overhead bar using an overhand grip (palms facing away from the body) with their arms fully extended, a pull-up was recorded when they use their arms to pull their bodies up until the chin was above the top of the bar. The next pull-ups required a starting position with extended arms. For sit-ups, female students were required to lie in a supine position with knees bent, feet flat on a floor mat (secured by the test examiner), hands placed on the back of the head. Then they were instructed to elevate their trunk until elbows touched thighs and to then return to the starting position by lowering their shoulder blades to the floor mat. In addition, male and female participants were required to complete the 1000 m and 800 m run test, respectively. Participants were instructed to run as fast as they could along a track. The time to finish this test was recorded.

In the end, the standardized values (z-score) of each test were calculated using the mean value of each test minus each students’ value, then the difference value was divided by the standard deviation of each test. A physical fitness index (defined as the total summed score of z-score of each test) was calculated for male, females, and the whole sample. Z-scores of the 50 m sprint and 1000 m/800 m run were reversed before the summing because a shorter time represented better performances. A muscular fitness index (calculated as the sum score of the standardized values (z-score) of standing long jump and sit-ups/pull-ups) were also generated for males, females, and the whole sample in this study.

### 2.4. Statistic Analysis

Stata MP 16.0 was used for statistical analysis. All continuous variables were expressed as means ± standard deviation (SD). The Kolmogorov–Smirnov tests were used for normality of distribution tests. The mean difference of continuous variables among groups was compared by the one-way analysis of variance (ANOVA). Further comparison was performed with the Bonferroni post-hoc test when statistical significance was observed by ANOVA. All categorical variables were presented by frequencies and percentages. A chi-square test was used for difference comparison among groups while the rank sum test was conducted for post-hoc multiple comparisons. Physical fitness was standardized and the sum of z-scores of each physical fitness component, to obtain a physical fitness index, was calculated. In addition, a muscular fitness index was calculated by the sum of z-scores of two muscular indicators. Three models were used to evaluate the potential relationship between BMI and physical fitness z-scores, BMI and muscular fitness z-scores, in both the total group and subgroups: (a) a linear regression model with BMI as categorical predictor, (b) a polynomial regression model with BMI as continuous predictor, and (c) a restricted cubic spline regression model with four knots. We compared the models using R-square. Statistical significance was set at a *p* value < 0.05.

## 3. Results

### 3.1. Basic Characteristics of Participants

The basic characteristic of participants grouped by year, weight status, and gender are listed in Table 1. A total of 6425 college freshmen who finished all physical fitness tests were enrolled in the final analysis. Most students were between 18 and 20 years old and of a normal weight; only a small number and proportion of college students were younger than 17 or older than 21 years old.

Age distribution differences among the four enrollment years were found in both total and gender subgroups (*p* < 0.001). Specifically, age distribution differences between the year 2017 to the three other years were all observed in total group (*p* < 0.001), male (*p* < 0.001), and female groups (*p* < 0.001), respectively. In the male group, the age distribution difference between the years 2018 to 2019 was also found to be significant (*p* = 0.046). However, weight status difference among the four enrollment years was just observed in the total group (*p* = 0.007), showing a significant difference between the year 2018 and the other three years (*p* = 0.049, *p* = 0.001, *p* = 0.031), years 2019 to 2020 (*p* = 0.038).

Difference in physical fitness score among the four enrollment years were also found in both total (*p* = 0.018) and the gender subgroups (*p* < 0.001), showing a significant difference between years 2019 to 2020 in all three groups (*p* = 0.047, *p* < 0.001, *p* < 0.001), years 2017 to 2019 in total group (*p* = 0.022) and male group (*p* < 0.023), year 2017 to 2020 in male (*p* < 0.001) and female group (*p* = 0.008), respectively. Difference of muscular fitness score among the four enrollment years were just found in male (*p* < 0.001) and female groups (*p* < 0.001) but not total group (*p* = 0.096), showing a significant difference between years 2017 to the other three years (*p* < 0.001, *p* = 0.008, *p* < 0.001) in male group, and years 2020 to other three years (*p* < 0.001, *p* < 0.001, *p* = 0.005) in female group. A significant difference in muscular fitness score between years 2019 and 2020 (*p* = 0.039) was also observed in the male group. It is notable that in the male group, a decreasing trend in both physical fitness score and muscular fitness score was observed with the increase/rise in freshmen year.

### 3.2. Analyses of Physical Fitness Stratified by BMI Category in College Students

We found a significant difference among different BMI categories in all physical fitness factors as well as in physical fitness z-score and muscular fitness z-score (*p* < 0.001), regardless of gender group (Table 2 and Table 3). The results of post hoc multiple comparison are as follows.

The performance in each single physical fitness test was significantly better for normal and overweight than the obesity group, regardless of gender (*p* < 0.05), except for sit-and-reach, which showed no difference between overweight and obesity group in female (*p* = 0.175) (Table 3). The performance in most single physical fitness tests in the underweight group was significantly better than the obesity group (*p* < 0.001) in both male and females, except for sit-and-reach in both male (Table 2) and female group (Table 3), and 1000-m run in male group (Table 2), respectively. The normal weight group performed significantly better in all single physical fitness tests than overweight group in both gender subgroups (*p* < 0.05), except for the sit-and-reach, which showed no difference in males (Table 2). The underweight group showed a significantly higher vital capacity weight index than the overweight group in both male and female groups (*p* < 0.001), while the overweight group had better performance in sit-and-reach than underweight in the male group (*p* = 0.047) (Table 2). In the female group, the underweight group performed better in the 50-m sprint (*p* < 0.001) and standing long jump (*p* = 0.001) than the overweight group (Table 3). When comparing the underweight to the normal weight group, both vital capacity weight index and sit-and-reach were found to be different between male and female subgroups (*p* < 0.05). In male group, significant differences were also observed in the 50-m sprint, 1000-m run, and pull-up, comparing the underweight to the normal weight group (*p* < 0.001) (Table 3).

For physical fitness z-score and muscular fitness z-score, a significantly lower value was found in obesity to other three BMI groups (*p* < 0.001), overweight to normal weight groups (*p* < 0.001), regardless of gender. The normal weight group had a significantly higher physical fitness z-score than underweight group among both male and females (*p* < 0.05). In addition, underweight females had a significant higher muscular fitness z-score than the overweight group (*p* = 0.004) (Table 3).

### 3.3. Regression Analysis of BMI Category and Muscular Fitness of College Freshmen

As Table 4 shows, significant linear associations were found between BMI value and physical fitness z-score as well as muscular fitness z-score among total, male, and female groups. Specifically, BMI value was significantly and negatively associated with physical fitness z-score in all three groups. Taking the normal weight group as a reference, underweight, overweight and obesity was associated with a significantly lower physical fitness z-score (1.04, 1.04, 2.96) in total group, male group (1.37, 1.06, 3.35), and female group (0.76, 1.05, 2.55). The association between BMI value and muscular fitness z-score was similar to its association with physical fitness z-score. Taking the normal weight group again as a reference, underweight, overweight and obesity was associated with a significantly lower muscular fitness z-score in the total group (0.30, 0.91, 2.08), male group (0.52, 1.10, 2.50), and female group (0.81, 1.67).

In addition, the restricted cubic spline regression model showed a better fitting effect (with an adjust R^2^ of 7.9%, 11.2%, and 4.8% in total, male, and female groups for physical fitness and 7.7%, 15.7%, and 4.0% for muscular fitness, respectively) compared with linear and polynomial regression model, presented by a higher adjusted R^2^ (Figure 1 and Figure 2). Restricted cubic splines analysis showed that BMI value and physical fitness z-score showed a non-linear relationship, with an approximate inverted U curve in all groups (Figure 1) while an approximate reversed J-shaped association was observed between BMI and muscular fitness z-score in all groups (Figure 2).

## 4. Discussion

The purpose of this study was to investigate the relationship between physical fitness, muscular fitness, and BMI. A sample of college freshmen was used for this cross-sectional study. Our results demonstrated a significant difference amongst different BMI categories in nearly all physical fitness components, as well as physical fitness z-score and muscular fitness z-score, regardless of gender. The restricted cubic spline regression model showed a better fitting effect compared with linear and polynomial regression models. Restricted cubic splines analysis showed that BMI value and physical fitness z-score showed a non-linear relationship with an approximate inverted U curve in all groups, while an approximate reversed J-shaped association was observed between BMI and muscular fitness z-score.

Our results showed that although the prevalence of overweight was not significantly changed, the prevalence of obesity exhibited an increasing trend in this college student population (6.98% in 2017 to 9.15% in 2020). Additionally, there was an apparent gender difference, with more overweight/obese male college students than female college students, which is consistent with previous studies conducted in China [25,26]. In addition, increasing evidence suggested that underweight is also associated with worse health outcomes, including increased all-cause mortality [27], higher risk of stroke and myocardial infarction [28], lower physical fitness [29,30], etc. We also found that the prevalence of underweight in college students showed a slight decrease (4.5% in 2017 vs. 3.7% in 2020) without significance. There was also no apparent gender difference in the prevalence of underweight between male college students (4.47% in 2017 to 4.09% in 2020) and female college students, which showed a slight decreasing trend (4.36% in 2017 to in 3.14% in 2020). This finding was different from previous studies, which reported a higher prevalence of underweight amongst female college students than male students [25,30]. In fact, both underweight and overweight Chinese students are increasing in recent years. A previous survey suggested that in 2009, about 24% of girls and 20% of boys aged 6 to 11 years old were underweight, and the underweight prevalence stayed high (around 20%) across the surveyed years in 12 to 18 year-old girls [31]. Usually, boys were especially prone to unhealthy lifestyles, leading to worsening body composition, and normal-to-overweight boys were more likely to believe themselves as underweight, making boys more susceptible to overweight or obesity [32]. On the contrary, girls are more likely to overestimate their weight and girls are more likely to perceive themselves as heavy, thus preferring to manage their weight or keep slim by restricting their diets. In addition, their parents still keep their stereotyped images of children—chubby boys and slim girls [33]. Portrayed body image in mass media also influences adolescent girls to think thinness is ideal and boys to desire muscular features [34]. Yet, a considerable number of boys desired a skinny figure and might have reduced their energy intake from regular meals while increasing soft drinks or snack intake. This unhealthy lifestyle might be continued to young adulthood. Thus, both overweight/obesity and underweight need efficacious political and educational interventions to encourage a healthy body image in Chinese youth and college students, involving a healthy diet and becoming more active and less sedentary.

In terms of the comparison of individual physical fitness components by BMI categories, a significant higher vital capacity performance in overweight and obesity groups was observed compared with normal weight and underweight college students, regardless of gender. However, when the vital capacity was averaged by weight, an opposite trend with overweight and obesity groups having the worst vital capacity weight index was found, which was in agreement with previous studies [25]. In addition, overweight/obese college students demonstrated a poorer performance in speed, lower body explosive strength, upper body muscular strength, endurance, abdominal muscular endurance, and aerobic endurance than normal weight college students, which was consistent with previous studies [25,35]. It is well-established that excess body weight and higher fat in the waist area leads to decreased abdominal muscular endurance and lower body explosive strength and lower cardiorespiratory fitness, which is in accordance with previous studies [29,36]. No significant difference in the performance of sit-and-reach between normal weight and overweight male college students was found. This study also suggested that overweight/obese college students had poorer performance in physical fitness z-score than normal weight groups, which was consistent with a study performed by Chen [25]. In addition, we also calculated the muscular z-score using standing long jump and pull-up/sit-up and produced similar results as the physical fitness z-score. By far, nearly all studies evaluated the muscular fitness by individual components including grip strength, pull-up/sit-up, push-up, and curl-up, rather than a standard score of two or more of these indicators [16,29]. This obesity-related worsening physical fitness performance might be associated with less physical activity participation due to weight stigma. Weight-related discrimination, both in sport and non-sport settings, is experienced in adolescence and adulthood [37]. People with obesity often cope with such experiences by simply excluding themselves from sport and exercise [38].

This study also suggested that underweight college students had poorer performance in physical fitness z-score, which was consistent with previous report by Chen [25]. There was an apparent gender difference in terms of individual physical fitness components. Underweight male college students had poorer performance in speed, flexibility, aerobic endurance, upper body muscle strength, and endurance than the normal weight group, while no significant difference was observed in lower body explosive strength and muscular fitness z-score. This also suggest that the lower body explosive strength contributed more to the muscular fitness. However, underweight female college students had similar performance in speed, lower body explosive strength, aerobic endurance, abdominal muscular endurance, but significant lower flexibility performance compared with normal weight. This result was different from previous studies which reported similar lower body explosive strength and pull-up performance in male students, and similar aerobic endurance and abdominal muscular endurance as well as the physical fitness z-score in female students [25,30]. This might be associated with the difference of underweight percentages in the sample as well as the geographic disparity and physical activity patterns.

Although consistent with previous studies [25,39], a significant linear association between BMI and physical fitness z-score as well as muscular fitness z-score among total, male, and female groups was observed, our results suggested that the relationship between BMI and physical fitness was non-linear with a reverse U-shaped relationship, which was consistent with a previous study by Huang [40]. Most previous studies reported an inverted J-shaped relationship between BMI and individual physical fitness components as well as physical fitness z-score [25,39]. The difference might be associated with a higher prevalence of underweight individuals in this study, which may better reveal the association between BMI and physical fitness. However, the relationship between BMI and muscular fitness was more likely to present an inverted J-shaped association in total, male, and female groups. It is easy to understand this phenomenon as the muscular fitness z-score calculated in this study used standing long jump and pull-up/sit-up which was negatively affected by weight. In addition, most of previous studies only analyzed the association between BMI and individual muscular power [41], muscular strength [29], and/or muscular endurance [29,39] rather than a standardized muscular fitness, and reported an inverted J-shaped relationship. The spine-restricted cubic splines (RCS) regression model is clearly superior to the linear and polynomial regression model for interpreting the association between BMI and physical fitness as well as muscular fitness.

There are several limitations to the present study. Firstly, this single-centered study cannot truly reflect the situation of the entire college freshmen population in China; a multi-centered study with a larger sample size is required in future. Secondly, physical fitness is affected by many factors; more variables, including physical activity, nutrition, and socioeconomic status should be considered in future studies. Thirdly, the physical fitness test followed principles of the national student fitness test of China, which is different from fitness tests performed in Western countries. Finally, the indicators and methods used in the present study might not truly reflect the actual muscular fitness level, more muscular fitness indicators like grip strength and push-ups should be considered.

## 5. Conclusions

This study revealed the trends of weight status of Chinese college freshmen and found a non-linear negative association between BMI and physical fitness z-score, characterized by a reverse U curve, and muscular fitness, characterized by an approximate inverted J-shape curve. Our findings may provide useful evidence for the development of public health recommendations and encourage the health management of young adults. Future multi-centered large sample size studies are required to test and verify the relationship and longitudinal cohort studies are required to identify the causal relation.

## Figures and Tables

**Figure 1 ijerph-19-14060-f001:**
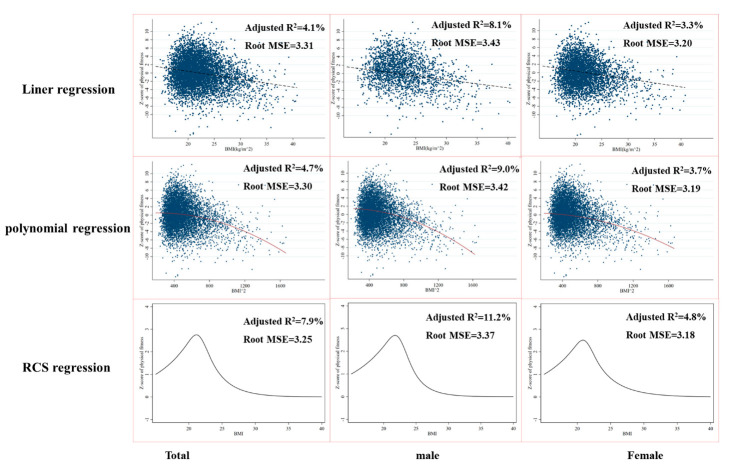
Linear regression, polynomial regression, and restricted cubic spines (RCS) analysis for BMI values and physical fitness z-score in college freshmen.

**Figure 2 ijerph-19-14060-f002:**
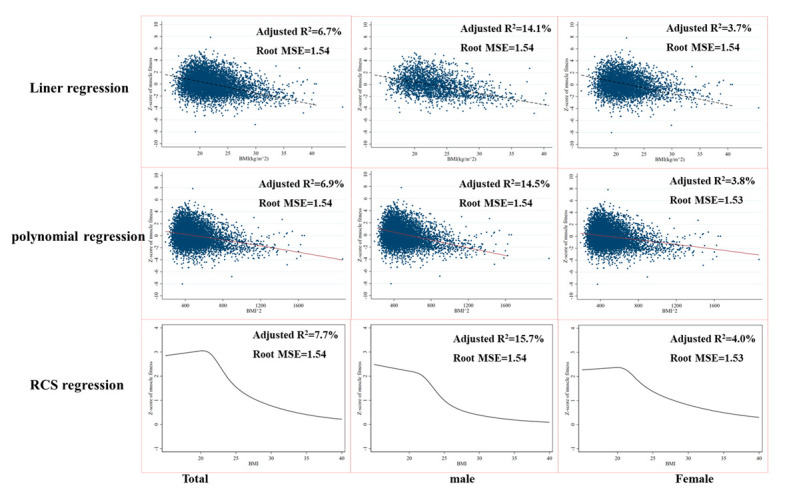
Linear regression, polynomial regression, and restricted cubic spines (RCS) analysis for BMI values and muscular fitness z-score in college freshmen.

**Table 1 ijerph-19-14060-t001:** The basic characteristics of participants.

Variables	Total				Male				Female			
	2017 (*n* = 1935)	2018 (*n* = 1681)	2019 (*n* = 1377)	2020 (*n* = 1432)	2017 (*n* = 696)	2018 (*n* = 481)	2019 (*n* = 453)	2020 (*n* = 521)	2017 (*n* = 1239)	2018 (*n* = 1200)	2019 (*n* = 924)	2020 (*n* = 911)
Age (year) [*n* (%)]												
≤17	70 (3.61)	64 (3.81)	72 (5.23)	56 (3.91)	18 (2.59)	14 (2.91)	14 (0.39)	14 (2.69)	52 (4.20)	50 (4.17)	58 (6.28)	42 (4.61)
18	1030 (53.23)	1070 (63.65)	855 (62.09)	913 (63.76)	336 (48.28)	293 (60.91)	277 (61.15)	334 (64.11)	694 (56.01)	777 (64.75)	578 (62.55)	579 (63.56)
19	643 (33.23)	470 (27.96)	397 (27.58)	395 (27.58)	248 (35.63)	148 (30.77)	144 (31.79)	153 (29.37)	395 (31.88)	322 (26.83)	253 (27.38)	245 (26.56)
20	160 (8.27)	69 (4.10)	48 (3.49)	58 (4.05)	78 (11.21)	23 (4.78)	17 (3.75)	15 (2.88)	82 (6.62)	46 (3.83)	41 (3.35)	43 (4.72)
21	24 (1.24)	7 (0.42)	3 (0.22)	8 (0.56)	13 (1.87)	2 (0.42)	1 (0.22)	3 (0.58)	11 (0.89)	5 (0.42)	2 (0.22)	5 (0.55)
≥22	8 (0.24)	1 (0.06)	2 (0.15)	2 (0.14)	3 (0.43)	1 (0.21)	0	2 (0.38)	5 (0.4)	0	2 (0.22)	0
X^2^ (df;*p*)	113.501 (15; *p* < 0.001)				77.929 (15; *p* < 0.001)				50.9 (15; *p* < 0.001)			
Post-hoc	P1 < 0.001; P2 < 0.001; P3 < 0.001; P4 = 0.550; P5 = 0.912; P6 = 0.638				P1 < 0.001; P2 < 0.001; P3 < 0.001; P4 = 0.046; P5 = 0.091; P6 = 0.197				P1 < 0.001; P2 < 0.001; P3 < 0.001; P4 = 0.756; P5 = 0.437; P6 = 0.151			
Weight status [*n* (%)]												
Underweight	87 (4.50)	54 (3.21)	54 (3.92)	53 (3.7)	33 (4.47)	26 (5.41)	25 (5.52)	26 (4.09)	54 (4.36)	28 (2.33)	29 (3.14)	27 (2.96)
Normal	1366 (70.59)	1293 (73.71)	944 (68.55)	1013 (70.74)	430 (61.33)	295 (61.33)	249 (54.97)	319 (61.23)	936 (61.23)	944 (78.67)	695 (75.22)	694 (76.18)
Overweight	347 (17.93)	287 (17.07)	265 (19.24)	235 (16.41)	162 (23.28)	114 (23.7)	118 (26.05)	104 (19.96)	185 (14.93)	173 (14.42)	147 (15.91)	131 (14.38)
Obesity	135 (6.98)	101 (6.01)	114 (8.28)	131 (9.15)	71 (10.2)	46 (9.56)	61 (13.82)	72 (13.82)	64 (5.17)	55 (4.58)	53 (5.47)	59 (6.48)
X^2^ (df;*p*)	22.562 (9; *p* = 0.007)				13.427 (9; *p* = 0.144)				13.932 (9; *p* = 0.125)			
Post-hoc	P1 = 0.049; P2 = 0.138; P3 = 0.721; P4 = 0.001; P5 = 0.031; P6 = 0.038				\				\			
PF z-score	0.12 (3.32)	0.009 (3.23)	−0.23 (3.38)	0.11 (3.62)	0.88 (3.48)	0.43 (3.38)	0.26 (3.41)	0.05 (3.97)	−0.31 (3.15)	−0.16 (3.16)	−0.47 (3.33)	0.51 (3.41)
F (df1; df2; *p*)	3.373 (3;6380; *p* = 0.018)				5.97 (3;21.29; *p* < 0.001)				6.05 (3; 4247; *p* < 0.001)			
Post-hoc	P1 = 1.000; P2 = 0.022; P3 = 1.000; P4 = 0.315; P5 = 1.000; P6 = 0.047				P1 = 0.203; P2 = 0.023; P3 < 0.001; P4 = 1.000; P5 = 0.554; P6 < 0.001				P1 = 1.000; P2 = 1.000; P3 = 0.008; P4 = 0.180; P5 = 0.197; P6 < 0.001			
MF z-score	0.05 (1.54)	−0.08 (1.55)	−0.01 (1.63)	0.04 (1.70)	0.31 (1.66)	−0.11 (1.53)	−0.01 (1.59)	−0.30 (1.77)	−0.06 (2.10)	−0.004 (2.18)	−0.12 (2.30)	0.22 (2.32)
F (df1; df2; *p*)	2.12 (3; 6421; *p* = 0.096)				14.64 (3; 2147; *p* < 0.001)				8.63 (3; 4270; *p* < 0.001)			
Post-hoc	\				P1 < 0.001; P2 = 0.008; P3 < 0.001; P4 = 1.000; P5 = 0.451; P6 = 0.039				P1 = 1.000; P2 = 1.000; P3 < 0.001; P4 = 1.000; P5 < 0.001; P6 = 0.005			

Notes: PF, physical fitness; MF, muscular fitness. Physical fitness score and muscular fitness score are described as mean (standard deviation). P1 represents 2017 versus 2018; P2 represents 2017 versus 2019; P3 represents 2017 versus 2020; P4 represents 2018 versus 2019; P5 represents 2018 versus 2020; P6 represents 2019 versus 2020.

**Table 2 ijerph-19-14060-t002:** Difference of physical fitness stratified by BMI category in male college students.

	Total	Underweight (a)	Normal Weight (b)	Overweight (c)	Obesity (d)	F	*p*	Post Hoc Multiple Comparison					
								a vs. b	a vs. c	a vs. d	b vs. c	b vs. d	c vs. d
Vital capacity	3949.3 ± 755.9	3531.6 ± 595.9	3879.7 ± 724.1	4100.7 ± 779.3	4191.6 ± 799.1	31.35	<0.001	<0.001	<0.001	<0.001	<0.001	<0.001	0.681
Vital capacity weight index	58.3 ± 12.1	68.3 ± 11.8	61.7 ± 11.1	53.5 ± 9.6	45.3 ± 8.3	241.42	<0.001	<0.001	<0.001	<0.001	<0.001	<0.001	<0.001
50-m sprint(s)	7.4 ± 0.5	7.5 ± 0.6	7.3 ± 0.5	7.5 ± 0.5	7.7 ± 0.5	51.9	<0.001	0.003	1.000	<0.001	<0.001	<0.001	<0.001
Sit-and-reach(cm)	13.8 ± 6.8	12.0 ± 7.0	14.2 ± 6.9	13.9 ± 6.9	12.1 ± 5.9	9.18	<0.001	0.008	0.047	1.000	1.000	<0.001	<0.003
Standing long jump(cm)	227.5 ± 19.5	227.8 ± 19.4	230.8 ± 18.8	224.8 ± 19.4	215.0 ± 18.0	50.35	<0.001	0.659	0.724	<0.001	<0.001	<0.001	<0.001
1000-m run(s)	251.7 ± 26.2	251.1 ± 20.9	247.4 ± 23.9	252.7 ± 23.3	272.5 ± 33.7	69.66	<0.001	<0.001	0.814	1.000	<0.001	<0.001	<0.001
Pull-up(numbers)	7.7 ± 5.5	8.6 ± 4.5	8.9 ± 5.5	6.2 ± 5.1	3.5 ± 4.0	93.69	<0.001	<0.001	1.000	<0.001	<0.001	<0.001	<0.001
Physical fitness z-score	0.5 ± 3.6	−0.3 ± 3.3	1.2 ± 3.3	0.1 ± 3.6	−2.3 ± 3.5	72.87	<0.001	<0.001	1.000	<0.001	<0.001	<0.001	<0.001
Muscular fitness z-score	0.0 ± 2.4	0.0 ± 2.3	0.6 ± 2.2	−0.5 ± 2.3	−1.9 ± 2.1	105.94	<0.001	0.075	0.105	<0.001	<0.001	<0.001	<0.001

**Table 3 ijerph-19-14060-t003:** Difference of physical fitness stratified by BMI category in female college students.

	Total	Underweight (a)	Normal weight (b)	Overweight (c)	Obesity (d)	F	*p*	Post Hoc Multiple Comparison					
								a vs. b	a vs. c	a vs. d	b vs. c	b vs. d	c vs. d
Vital capacity	2761.92 ± 514.88	2490.49 ± 468.06	2735.58 ± 501.92	2860.83 ± 535.76	3024.41 ± 521.8	44.81	<0.001	<0.001	<0.001	<0.001	<0.001	<0.001	<0.001
Vital capacity weight index	49.60 ± 9.90	57.26 ± 10.63	51.36 ± 9.17	43.41 ± 8.03	37.18 ± 6.48	326.40	<0.001	<0.001	<0.001	<0.001	<0.001	<0.001	<0.001
50-m sprint	9.12 ± 0.63	9.11 ± 0.71	9.07 ± 0.62	9.25 ± 0.62	9.42 ± 0.65	36.50	<0.001	0.088	<0.001	<0.001	<0.001	<0.001	<0.001
Sit-and-reach	18.98 ± 5.83	17.62 ± 6.12	19.24 ± 5.9	18.48 ± 5.51	17.50 ± 4.99	11.31	<0.001	0.008	0.708	1.000	0.014	<0.001	0.175
Standing long jump	169.05 ± 15.18	170.30 ± 15.13	170.28 ± 15.23	165.09 ± 14.20	161.65 ± 13.13	41.32	<0.001	1.000	0.001	<0.001	<0.001	<0.001	0.017
800-m run	242.18 ± 20.33	241.90 ± 18.98	240.14 ± 19.26	246.25 ± 19.83	260.53 ± 25.97	85.65	<0.001	1.000	0.116	<0.001	<0.001	<0.001	<0.001
Sit-up	37.51 ± 8.86	37.44 ± 8.76	38.02 ± 8.87	36.35 ± 8.53	33.46 ± 8.39	23.73	<0.001	1.000	1.000	<0.001	<0.001	<0.001	<0.001
Physical fitness z-score	−0.20 ± 3.25	−0.66 ± 3.14	0.11 ± 3.23	−0.95 ± 3.02	−2.43 ± 3.03	60.02	<0.001	0.031	1.000	<0.001	<0.001	<0.001	<0.001
Muscular fitness z-score	0.00 ± 2.22	0.09 ± 2.33	0.22 ± 2.20	−0.60 ± 2.04	−1.45 ± 2.03	61.64	<0.001	1.000	0.004	<0.001	<0.001	<0.001	<0.001

**Table 4 ijerph-19-14060-t004:** Linear regression analysis for physical fitness and weight status in college students.

Variables	BMI	Total			Male			Female		
		β	95%CI	*p*-Value	β	95% CI	*p*-Value	β	95% CI	*p*-Value
Physical fitness z-score	underweight	−1.04	−1.46, −0.62	<0.01	−1.37	−2.04, −0.71	0.006	−0.76	−1.31, −0.22	0.006
	overweight	−1.04	−1.25, −0.82	<0.01	−1.06	−1.41, −0.70	<0.01	−1.05	−1.33, −0.78	<0.01
	obesity	−2.96	−3.27, −2.64	<0.01	−3.35	−3.82, −2.88	<0.01	−2.55	−2.98, −2.12	<0.01
Muscular fitness z-score	underweight	−0.30	−0.58, −0.02	0.038	−0.52	−0.95, −0.10	0.016	−0.12	−0.49, 0.25	0.512
	overweight	−0.91	−1.06, −0.77	<0.01	−1.10	−1.32, −0.88	<0.01	−0.81	−0.10, −0.63	<0.01
	obesity	−2.08	−2.29, −1.87	<0.01	−2.50	−2.80, −2.21	<0.01	−1.67	−1.96, −1.38	<0.01

## Data Availability

Not applicable.

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
