# Peer review of "The Association between Body Mass Index and Muscular Fitness in Chinese College Freshmen"

_ijerph, 2022, doi:10.3390/ijerph192114060_

Round 1

Reviewer 1 Report

As attached notes

Author Response

Title

The use of abbreviations and other non-specific terms should be avoided.

Thanks for your suggestion. We have spelled out the abbreviation in full.

Abstract

1) It is written in a structured way, however, the methodology is written in a very summarized way which ends up making the findings and conclusions of the article.

2) There are some abbreviations, even if they are common, that are not presented in full the first time they appear in the text.

3) The values found are not included in the abstract, which makes it difficult to understand what was done.

4) Another issue is that concussions do not address practical applications of the findings.

5) Please confirm that Keywords are listed as descriptors in health sciences

Thanks for your suggestion.

1) We have made expansion of methodology;

2) We have spelled out the abbreviations in full the first time when they appeared in this abstract;

3) We have added the corresponding values in the abstract;

4) We have addressed practical applications of the findings in conclusion.

5) We have carefully considered and modified the keywords.

Introduction

This is very extensive, and on the other hand methodologically explains some points that should this in methodology and not in the introduction.

The introduction is not starting from general to specific. It should initially present a more general approach and gradually address the problem (gap) and then present the objective.

From what is written in the objectives and based on what the study proposes, the study evaluated the relationship between body mass index (BMI) and physical fitness performed through specific tests. However, this is not what is described in the introduction. The introduction should focus on the objective of the study and not on variables peripheral to the study.

The introduction should be more focused on the construct and not on the methodology of what is being researched.

It would be important for the manuscript to bring the hypotheses to be answered by the study.

Thanks for your suggestion. We have reorganized the structure of the introduction part.

Methods

1) It should present more clearly the design of the study.

2) A CONSORT or time line, should be presented in order to get a better view of the study design.

3) The sample should be better explained with the number of subjects presented initially and then present the inclusion and exclusion criteria.

4) Statistical treatment should be better detailed in order to better follow what has been done. Please consult Cohen (1988).

Thanks for your suggestion.

1) The study design was added.

2) The time point of the study was added.

3) The details of the sample and inclusion criteria were presented.

4) The statistical analysis part was modified with more details.

Results

The way in which the results were presented makes it difficult to visualize them. Ideally, references should be made to each of the tables and/or figures and presented and the main findings explained. This should occur in all results.

Thanks for your suggestion. We have added references of tables and figures in the corresponding places

Discussion

1) It should reaffirm the objectives and start discussing the results in the chronological order that appear in the item results.

Results should be discussed in the chronological order in which the results were presented.

Thanks for your suggestion. We have modified the discussion part in the chronological order as the result presented.

2) In the limitations several limitations are mentioned, but in the text only three limitations are presented.

Thanks for your suggestion. We have carefully considered this part and provided more descriptions.

Conclusion

Are presented satisfactorily. However, no practical applications of the findings were presented.

Thanks for your suggestion. We have added practical applications of the findings in conclusion part.

References

Of the 55 references, 32 are current and 23 are more than five years old. The formatting of references must be checked in accordance with the journal's rules. Another important point is that the authors mention that there are several studies on the topic, so it would be recommended that the references be updated.

Thanks for your suggestion. We have updated the references with more new articles published recently and the format of the references was checked again to make sure they followed ijerph’s rules.

Reviewer 2 Report

Author provided the most updated fitness profiles from the Chinese college age group which is very useful for exercise planning and further improvement.

The following items are suggested for further revision.

1.       Line 13, 106 Check and review with literature about pull-ups test should be muscular strength and/or endurance

2.       Line 14, 105 Check and review with literature about sit-ups test should be muscular strength and/or endurance

3.       Line 32 A very important topical sentence to support the rationale of the study. Need to cite literature reference indeed.

4.       Line 107 Model and production country of spirometer is missing

5.       Line 287 Need to clarify or providing reference support if vital capacity is belonging to the physical fitness component.

Author Response

1) Line 13, 106 Check and review with literature about pull-ups test should be muscular strength and/or endurance

Thanks for your suggestion. We have further checked and review literatures about pull-ups in physical fitness test and finally confirm that pull-ups was considered as an indicator of muscular endurance (PMID:30404195;30247275; 19826295).

2) Line 14, 105 Check and review with literature about sit-ups test should be muscular strength and/or endurance

Thanks for your suggestion. We have further checked and review literatures about sit-ups in physical fitness test and finally confirm that sit-ups was considered as an indicator of abdominal muscular endurance (PMID:19826295; 24146716; 32260379;20429737).

3) Line 32 A very important topical sentence to support the rationale of the study. Need to cite literature reference indeed.

Thanks for your suggestion. We have added a literature reference (PMID: 29029897) for this sentence.

4) Line 107 Model and production country of spirometer is missing

Thanks for your suggestion. We have added the detailed information about the model and production country of spirometer used in this study.

5) Line 287 Need to clarify or providing reference support if vital capacity is belonging to the physical fitness component.

Thanks for your suggestion. Vital capacity was enrolled in physical fitness test in many previous studies (PMID: 34501479, 33665022, 32197926, 24331683, etc.)

Reviewer 3 Report

This paper is an interesting read. The sample size is big with more than 6400 participants and the data show values of big importance related to the current health status of the Chinese population. There are some concerns that the authors must address before the publication of this work. The introduction and results sections are the strong points of this work but the material and method section is very poor compared to the rest of the paper.

Abstract

L11. Could you provide the values of the classification?

L15. You are referring to z-scores that you did not mention in the method section of the abstract. Please, describe these outcomes before showing their results.

Introduction

The introduction is well written and introduces both the seriousness of the current state of obesity in general and the Chinese population and the relationship with physical fitness. No changes are required in this section.

Material and methods

L92. ...were finally included for analysis.

L99. Are the authors sure that overweight considered by the WHO is a BMI between 24 and 28? I just checked it on the WHO website and overweight is considered for a BMI from 25 to 30, and obesity over 30. Authors must change these values and the analysis according to these BMI values or cite the document where the WHO gives those values of BMI.

L102-118. The tests are poorly explained. Authors only mention the tests performed with a vague description, they must describe how the tests were carried out with all the details.

L120-121. Also, it must be described how the z-scores were calculated.

L128. Why did you use the Shapiro-Wilk test to check the normality of the data instead of Kolmogorov-Smirnov with such a big sample?

Results.

Please, show a table with the sample size (n) of the male group, female and different groups depending on BMI to ease the interpretation of the results to the readers.

Author Response

Abstract

L11. Could you provide the values of the classification?

Thanks for your suggestion, we have added the specific values of BMI classification in the abstract.

L15. You are referring to z-scores that you did not mention in the method section of the abstract. Please, describe these outcomes before showing their results.

Thanks for your suggestion, we have added information about z-scores in the abstract.

Introduction

The introduction is well written and introduces both the seriousness of the current state of obesity in general and the Chinese population and the relationship with physical fitness. No changes are required in this section.

Thanks for your comments.

Material and methods

L92. ...were finally included for analysis.

Thanks for your suggestions. We have changed “was” to “were”.

L99. Are the authors sure that overweight considered by the WHO is a BMI between 24 and 28? I just checked it on the WHO website and overweight is considered for a BMI from 25 to 30, and obesity over 30. Authors must change these values and the analysis according to these BMI values or cite the document where the WHO gives those values of BMI.

Thanks for your question. Sorry for our carelessness. The criteria in the present study was based on Working Group on Obesity in China (PMID: 12046553).

L102-118. The tests are poorly explained. Authors only mention the tests performed with a vague description, they must describe how the tests were carried out with all the details.

Thanks for your suggestion, we have added testing details in the text.

L120-121. Also, it must be described how the z-scores were calculated.

Thanks for your suggestion, we have added calculating details of z-score in the text.

L128. Why did you use the Shapiro-Wilk test to check the normality of the data instead of Kolmogorov-Smirnov with such a big sample?

Thanks for your question. We have replaced Shapiro-Wilk test to Kolmogorov-Smirnov test, and the results remained unchanged.

Results.

Please, show a table with the sample size (n) of the male group, female and different groups depending on BMI to ease the interpretation of the results to the readers.

Thanks for your suggestion. The required data have already been shown in Table 1.

Round 2

Reviewer 1 Report

After the adaptations made by the authors, I consider the manuscript in conditions of publication.

Author Response

Thanks for your approval!

Reviewer 3 Report

Authors have addressed most of my concerns and the quality of the paper has improved significantly.

However, authors are using the BMI values established by the Working group on Obesity in China. As the papers published in this journal are read by researchers around the world, authors must used the offical BMI values marked by the WHO (<18; 18-25; 25-30; 30-35...) along the whole manuscript

Author Response

Thanks for your suggestions. The data used to derive WHO criteria were mainly from Caucasian populations[1]. A WHO expert consultation early acknowledged that the proportion of Asian people with a high risk of type 2 diabetes and cardiovascular disease was substantial at BMIs lower than the existing WHO cut-off point for overweight (BMI≥25 kg/m2)  compared to Caucasian populations[2]. This, at least in part, may result in similar health risks in Asians at lower levels of BMI than in Caucasian populations. Several countries such as China, Japan and Indian have adopted their own cutoff points for the classification of BMI. The Asia Pacific Cohort Studies Collaboration has proposed that the international classification of obesity should be adapted for Asian countries[3]. They indicated that, in Asian populations, overweight should be classified as a BMI above 23kg/m2 and obesity as a BMI of 25kg/m2 or higher. If such a classification was applied, the prevalence of obesity (BMI ≥25kg/m2) in Japan would be substantially higher (over 20% rather than 2-3%) [4]. In addition, the revised guidelines categorized overweight as a BMI of 23-24.9 kg/m2 and obesity as a BMI ≥25kg/m2 for Asian Indians[5]. The ethnicity-specific BMI categories for overweight and obesity are 24 kg/m2 and 28 kg/m2 respectively in China by Working Group on Obesity in China (WGOC), which was based on surveys on 239,972 people, aged 20–70 in 1990s covering 21 provinces[6]. The global prevalence of obesity may, therefore, be vastly underestimated because many people in Asia may be inappropriately classified by their level of BMI.

That’s why we used the WGOC recommended criteria for the classification of underweight, overweight and obesity in this study.

As a response, we did follow your suggestion and used the WHO criteria to reanalyze our original data and the results are added as an appendix. We found that there’s some difference (marked by yellow colour) in the results. Firstly, the weight status difference disappeared among four enrollment years in total group (P=0.065 when used WHO criteria vs P=0.007 when used WGOC criteria) as Table S1 listed.

Secondly, the mean difference of some physical fitness components (Sit-and- reach, 1000-m runs, pull-ups) between different BMI categories in male students significantly changed after using the WHO criteria. Specifically, in terms of sit-and-reach, the difference between underweight and overweight male students becomes nonsignificant while the difference between normal weight and overweight becomes significant. In terms of 1000-m run, the difference between underweight and normal weight becomes nonsignificant while the difference between underweight and obesity becomes significant. In terms of pull-ups, the difference between underweight and normal weight becomes nonsignificant while the difference between underweight and overweight becomes significant.

Finally, the linear regression analysis suggested that the association between BMI and physical fitness-Z score as well as muscular fitness-z score becomes confused after using the WHO criteria. Specifically, the significant negative association between underweight and physical fitness-Z score in female group with normal weight category as reference disappeared although still presenting a trend of negative relationship. Similarly, the significant negative association between underweight and muscular fitness-Z score in total and male group disappeared. In addition, the β values become smaller in overweight and obesity category.  However, these changes do not influence the Linear regression, polynomial regression, and restricted cubic spines (RCS) analysis results as BMI was used as a continuous variable.

So we would like to insist on using the WGOC recommend criteria and please consider our suggestions as WHO criteria might be not appropriate for Chinese adults. 

References

[1] Obesity: preventing and managing the global epidemic. Report of a WHO consultation[J]. World Health Organ Tech Rep Ser. 2000, 894: 1-253.

[2] Appropriate body-mass index for Asian populations and its implications for policy and intervention strategies[J]. Lancet. 2004, 363(9403): 157-163.

[3] The burden of overweight and obesity in the Asia-Pacific region[J]. Obes Rev. 2007, 8(3): 191-196.

[4] New criteria for 'obesity disease' in Japan[J]. Circ J. 2002, 66(11): 987-992.

[5] Misra A, Chowbey P, Makkar B M, et al. Consensus statement for diagnosis of obesity, abdominal obesity and the metabolic syndrome for Asian Indians and recommendations for physical activity, medical and surgical management[J]. J Assoc Physicians India. 2009, 57: 163-170.

[6] Chen C, Lu F C. The guidelines for prevention and control of overweight and obesity in Chinese adults[J]. Biomed Environ Sci. 2004, 17 Suppl: 1-36.